# The Extracellular Molecular Chaperone Clusterin Inhibits Amyloid Fibril Formation and Suppresses Cytotoxicity Associated with Semen-Derived Enhancer of Virus Infection (SEVI)

**DOI:** 10.3390/cells11203259

**Published:** 2022-10-17

**Authors:** Abigail K. Elias, Mark R. Wilson, John A. Carver, Ian F. Musgrave

**Affiliations:** 1School of Physical Sciences, The University of Adelaide, Adelaide, SA 5005, Australia; 2Department of Hematology, Liverpool Hospital, New South Wales Pathology, Liverpool, NSW 2170, Australia; 3School of Chemistry and Molecular Bioscience, University of Wollongong, Wollongong, NSW 2522, Australia; 4Molecular Horizons Research Institute, University of Wollongong, Wollongong, NSW 2522, Australia; 5Illawarra Health and Medical Research Institute, University of Wollongong, Wollongong, NSW 2522, Australia; 6Research School of Chemistry, The Australian National University, Acton, Canberra, ACT 2601, Australia; 7School of Biomedicine, The University of Adelaide, Adelaide, SA 5005, Australia

**Keywords:** protein aggregation, SEVI, clusterin, amyloid fibril, cytotoxicity, molecular chaperone

## Abstract

Clusterin is a glycoprotein present at high concentrations in many extracellular fluids, including semen. Its increased expression accompanies disorders associated with extracellular amyloid fibril accumulation such as Alzheimer’s disease. Clusterin is an extracellular molecular chaperone which prevents the misfolding and amorphous and amyloid fibrillar aggregation of a wide variety of unfolding proteins. In semen, amyloid fibrils formed from a 39-amino acid fragment of prostatic acid phosphatase, termed Semen-derived Enhancer of Virus Infection (SEVI), potentiate HIV infectivity. In this study, clusterin potently inhibited the in vitro formation of SEVI fibrils, along with dissociating them. Furthermore, clusterin reduced the toxicity of SEVI to pheochromocytoma-12 cells. In semen, clusterin may play an important role in preventing SEVI amyloid fibril formation, in dissociating SEVI fibrils and in mitigating their enhancement of HIV infection.

## 1. Introduction

Numerous debilitating and incurable human disorders are associated with extracellular depositions of protein aggregates in a variety of organs and tissues, including the brain (e.g., Alzheimer’s disease), skeletal tissues and joints (e.g., haemodialysis-related amyloidosis) and the liver, spleen and heart (e.g., systemic amyloidosis) [1,2,3]. The protein aggregates are rope-like, highly structured, β-sheet-containing entities termed amyloid fibrils. They arise when a specific protein or protein fragment loses its native conformation (whether that be folded or unstructured), subsequently aggregates and forms insoluble deposits. The trigger for misfolding may be stresses such as elevated temperature, low pH, infection, the presence of reactive oxygen species or inherited mutations [4]. The intermediately folded proteins that result expose greater hydrophobic region(s) to solution which encourages their misfolding, mutual association and hence aggregation [3]. 

Proteostasis is a term that describes the various mechanisms by which correct protein levels in vivo are regulated intra- and extracellularly, including the inhibition of inappropriate protein aggregation. Proteostatic mechanisms include protein quality control and degradation processes and the production of molecular chaperones. Molecular chaperones are a large group of proteins whose function is to interact with destabilized proteins to prevent protein aggregation and, in some cases, facilitate proteins to fold into their correct structure. The majority of known molecular chaperones are found intracellularly. However, many amyloid fibrillar aggregates accumulate in extracellular space [3,5]. Extracellular molecular chaperones have been identified; they include clusterin [3,6], haptoglobin [7], α_2_-macroglobulin [8] and α_S_- and β-casein [9,10]. Recently, neuroserpin, transthyretin, vitronectin and plasminogen activator inhibitor-3 have been identified as extracellular chaperones [11,12]. The first-described and most well-characterized extracellular chaperone is clusterin [3,5,6]. 

Clusterin is a glycoprotein found in a variety of tissues whose expression shows tissue-specific patterns [13]. Clusterin is present at relatively high levels in plasma (0.035–0.105 mg/mL) [14] and semen (0.5–3.5 mg/mL) [15]. In vivo, clusterin is bound to many disparate proteins. As a result, it has been ascribed a diverse range of functions from being a regulator of apoptosis to a lipid transporter and a regulator of complement [3,16]. Elevated clusterin levels in serum are associated with rapid clinical progression in Alzheimer’s disease and accordingly, it has been identified as a candidate biomarker of the disease [17]. We have shown that clusterin has chaperone activity with the ability to prevent the amorphous and amyloid fibrillar aggregation of a broad range of proteins [2,3,5,6,18,19]. Clusterin stabilizes amorphously aggregating proteins under stress conditions, e.g., elevated temperature, by sequestering them into soluble high-molecular-weight complexes and interacts transiently with amyloid fibril-forming proteins to prevent their aggregation [20]. 

Previously, we demonstrated that fibrils formed by a 39-amino acid fragment of Prostatic Acid Phosphatase (PAP248–286), referred to as Semen-derived Enhancer of Virus Infection (SEVI), are toxic to pheochromocytoma (PC)-12 cells [21]. When present, SEVI enhances HIV infection by up to five orders of magnitude by assisting HIV to attach to cells [22]. The concentration of SEVI in semen can be high as the amount of PAP248–286 produced in semen is approximately 1–2 mg/mL [22]. In aqueous solution, PAP248–286 adopts an unstructured, intrinsically disordered conformation, i.e., it exhibits no preferred secondary structure [21,23]. In the presence of SDS micelles, however, PAP248–286 adopts a partial (albeit nascent) helical conformation in two short regions of the peptide, the longer one being in the centre of the peptide [24]. In addition, PAP248–286 interacts weakly with the surface of the micelle [24]. In the presence of the membrane-mimicking solvent, 50% *v/v* trifluoroethanol/water, however, PAP248–286 adopts an amphipathic α-helical structure along most of its length that is very similar to its conformation in intact prostatic acid phosphatase [23]. Contrastingly, PAP248–286 undergoes a major structural rearrangement upon aggregation and conversion to an amyloid fibrillar structure to form SEVI, with its polypeptide chain arranged in a highly ordered, cross β-sheet array [22].

In this study, we investigated whether clusterin (i) inhibits the formation of SEVI amyloid fibrils in vitro, (ii) is capable of dissociating SEVI fibrils, and (iii) prevents SEVI fibril-associated cytotoxicity.

## 2. Materials and Methods

### 2.1. SEVI Amyloid Fibril Formation

PAP248–286 peptide (>95% purity by HPLC) was custom synthesized by Mimotopes, The Peptide Company, Melbourne, Australia. For aggregation studies, PAP248–286 (2 mg/mL; 439 μM) was dissolved in 200 mM phosphate buffer, pH 7.2. Clusterin was also dissolved in the same phosphate buffer and added to PAP248–286 at 0, 3, 20, 40 and 200 µg/mL; 0, 0.050, 0.333, 0.666 and 3.33 μM clusterin monomer respectively). Samples were incubated at 37 °C with shaking (700 rpm) for two days to allow fibril formation. Samples were removed from solution every 8 h and snap frozen at −20 °C.

### 2.2. Clusterin

Clusterin was isolated from human blood serum, as outlined previously [5,6].

### 2.3. Thioflavin T Fluorescence Assay

Samples that were frozen as outlined above were defrosted and 10 µL of each sample was added to 25 µL of thioflavin T (ThT) (0.5 mM in phosphate buffer) and made to a final volume of 200 µL with 200 mM phosphate buffer, pH 7.2. The fluorescence intensity of each sample was measured with excitation and emission wavelengths set at 440 nm and 490 nm, respectively, in triplicate wells of a 96 well plate using FLUOstar and POLARstar Optima microplate readers (BMG Labtechnologies, Melbourne, Australia). 

### 2.4. Transmission Electron Microscopy

2 µL of aggregate samples from the ThT assays was added to formavar and carbon-coated nickel grids (SPI supplies, West Chester, PA, USA) for 2 min. The grids were washed three times with 10 µL of water and negatively stained with 10 µL uranyl acetate (2% *w/v*). Excess stain was removed with filter paper and the grids were air dried and viewed using a Tecnai G2 Spirit transmission electron microscope (Philips, Eindhoven, The Netherlands) at 10,000× magnification.

### 2.5. Cellular Toxicity

Pheochromocytoma (PC)-12 cells were grown in Dulbecco’s Modified Eagle’s Medium (DMEM, Gibco, Victoria, Australia) containing 5% *v/v* fetal calf serum, 1% *w/v* glutamate, non-essential amino acids and penicillin and streptomycin in 75 mL flasks at 37 °C in an incubator with 95% air and 5% carbon dioxide. Cells were passaged every 2–3 days into fresh medium. The cells were plated into a 96 well plate at a density of 2 × 10^4^ cells/well and incubated for 24 h. The cells were then treated with samples of SEVI with varying clusterin concentrations (six replicates per treatment) and incubated for 48 h. Cell viability was determined by the 3-(4,5-dimethylthiazol-2-yl)-2,5-diphenyltetrazolium bromide (MTT) assay.

### 2.6. MTT Assay

A 0.60 mM solution of MTT was prepared (3 mg of MTT (Sigma Aldrich, Sydney, Australia) dissolved in 12 mL of serum-free DMEM). The media was aspirated from the 96 well plate and 100 µL of MTT-containing media was added to each well. The plate was incubated for three hours at 37 °C. The MTT-containing media was aspirated and 100 µL of DMSO was added to each well. Formazan absorption was measured at 560 nm using a BMG Polarstar microplate reader (BMG Labtechnologies, Offenburg, Germany). The mean of the absorbances of the six replicates per treatment was taken in each 96 well plate, and cell viability was calculated by dividing the average absorption readings of treated wells with average absorption readings of the six replicate untreated wells (phosphate buffer only). Three independent experiments were conducted.

## 3. Results

### 3.1. The Effect of Clusterin on the Aggregation and Amyloid Fibril Formation of PAP248–286

In vitro, clusterin potently inhibits amyloid fibril formation of a variety of peptides and proteins including the Amyloid-β (Aβ) peptide [25,26,27], a fragment of prion protein, PrP (106–126) [28], and apolipoprotein C-II (apoC-II) [29]. We sought to determine whether the chaperone (anti-aggregation) activity of clusterin also applies to the inhibition of PAP248–286 aggregation and SEVI fibril formation. A ThT fluorescence assay was used to monitor the aggregation of 2 mg/mL PAP248–286 in the presence of varying concentrations of clusterin at 37 °C for 48 h. In the presence of increasing clusterin, a concentration-dependent decrease in SEVI-associated ThT fluorescence occurred (Figure 1). After 48 h of incubation, 3 µg/mL clusterin decreased the ThT fluorescence to 54 ± 4% of its value in the absence of clusterin, while 200 µg/mL clusterin reduced ThT fluorescence levels to background, implying complete inhibition of fibril formation. Under these conditions, clusterin potently inhibited PAP248–286 aggregation at a molar ratio of 132:1 PAP248–286:clusterin monomer. Clusterin exhibits similar sub-stoichiometric ability in preventing the aggregation of a variety of amorphous and fibril-forming target proteins [5,6,13]. 

To determine whether clusterin could disaggregate SEVI fibrils, 200 µg/mL of clusterin was added to preformed 2 mg/mL SEVI fibrils and incubated at 37 °C with samples taken every 8 h over a 48 h period. After this time, ThT fluorescence decreased by 77 ± 3% of its initial value implying a significant reduction in the presence of amyloid fibrils. Furthermore, clusterin incubated alone did not change its ThT fluorescence with time, indicating that clusterin does not form fibrils under these conditions, as observed previously [29]. 

### 3.2. Transmission Electron Microscopic Analysis of the Effect of Clusterin on SEVI Amyloid Fibril Formation

The effect of clusterin on SEVI fibril morphology (using samples from the ThT assays after 48 h of incubation) was assessed by TEM (Figure 2). In accordance with the ThT assay, the SEVI sample formed in the absence of clusterin exhibited long fibrillar aggregates of dimensions characteristic of amyloid fibrils (Figure 2a). The fibrils were of very similar density and morphology to those observed previously for SEVI [21]. The morphology of SEVI fibrils formed in the presence of 3 µg/mL clusterin (Figure 2b) was similar to the dense, long fibrils of SEVI formed in the absence of clusterin. In contrast, SEVI formed in the presence of clusterin at 20 µg/mL (Figure 2c) and 40 µg/mL (Figure 2d) produced scattered short fibrils, the latter shorter than the former. At 200 µg/mL clusterin (Figure 2e), amorphous aggregates, not fibrils, were observed. 

As mentioned above, in vitro clusterin suppresses amyloid fibril formation of a broad range of peptides and proteins [2,5,26,27,28,29,30]. In doing so, clusterin interacts with prefibrillar species rather than the monomeric peptide/protein or mature fibrils [5,13]. Hatters et al. [29] proposed that clusterin inhibits apoC-ll amyloid fibril formation by interacting stoichiometrically with amyloidogenic precursors (nuclei) of apoC-II, leading to dissociation of the nuclei back to monomer thereby inhibiting fibril growth. Using single molecule techniques, Narayan et al. [31] showed that clusterin interacts with a range of oligomeric forms of Aβ thereby inhibiting further growth to fibrils, or dissociates the oligomers to monomers. 

In this study, the highest concentration of clusterin tested (200 µg/mL), i.e., a PAP248–286:clusterin monomer ratio of 132:1, completely inhibited the formation of SEVI fibrils, as measured by both ThT fluorescence and TEM. At lower concentrations of clusterin (20 and 40 µg/mL), clusterin reduced fibril formation (as assessed by the ThT assay), and those fibrils that were formed were shorter in length (as monitored by TEM). The TEM images of the effect of 200 µg/mL clusterin on preformed SEVI fibrils were similar to those in which SEVI was formed in the presence of 40 µg/mL clusterin (compare Figure 2d,f, i.e., some fibrillar species, of smaller size, than those formed in the absence of clusterin (Figure 2a) were present. As the preformed fibrils incubated with clusterin appeared to undergo significant dissociation, the effects of a longer incubation (three days) and a higher concentration of clusterin (400 µg/mL) on SEVI fibril morphology was examined. After three days of incubation of SEVI with 200 µg/mL clusterin, no further dissociation of fibrils was noted. Furthermore, higher concentrations of clusterin (400 µg/mL) dissociated the fibrils but did not remove all of them (results not shown). As a control, a 2 mg/mL sample of preformed SEVI fibrils incubated for three days without clusterin (Figure 2g) exhibited significantly more fibrils than in the presence of clusterin (Figure 2f). 

Thus, the combined results of ThT and TEM analysis show that, at impressive sub-stoichiometric levels, clusterin prevents fibril formation of SEVI. In addition, clusterin dissociates preformed SEVI fibrils to smaller species. The ability of clusterin to prevent SEVI fibril formation at a sub-stoichiometric level is consistent with the notion that clusterin is a potent generic inhibitor of amyloid fibril and amorphous aggregation [2,3,5,6,29]. The TEM images indicate that the chaperone action of clusterin at higher concentration leads to the formation amorphous aggregates of PAP248–286 (Figure 2e).

### 3.3. Clusterin Inhibits the Cytotoxicity of SEVI 

Several studies have shown that clusterin either enhances [27,30] or suppresses [3] the cytotoxicity of Aβ but each of the studies used different conditions. Since human semen contains relatively high concentrations of both clusterin and PAP248–286, we investigated the effect of clusterin on the cytotoxicity of SEVI fibrils. Samples from the ThT assay, after 48 h of incubation, were added to PC-12 cells and the toxicity of samples with clusterin was compared to that of SEVI without clusterin present. SEVI (2 mg/mL) caused a reduction in cell viability of 34 ± 4% while PAP248–286 samples incubated with 3 and 20 µg/mL of clusterin had no significant alteration in cell viability compared to SEVI (30 ± 2% and 31 ± 6% reduction, respectively, Figure 3). 40 µg/mL of clusterin provided some protection with cell viability reduction of 25 ± 6%, whereas total suppression of SEVI toxicity was observed at 200 µg/mL of clusterin (108 ± 9% viability of control, *p* < 0.01, Figure 3). Clusterin (200 µg/mL) added to preformed SEVI fibrils (which are toxic to cells) had no significant impact on cytotoxicity (28 ± 6% reduction, Figure 3), despite being able to dissociate the long fibrillar species into smaller fibrillar species (as demonstrated in the TEM images in Figure 2f). However, the ThT-monitored dissociation, TEM appearance and degree of cell death were similar for experiments in which 40 µg/mL clusterin was added simultaneously and 200 µg/mL clusterin was added to preformed fibrils (25 ± 6% versus 28 ± 6% reduction, respectively; Figure 1 and Figure 3, and compare Figure 2d,f). Thus, it is not clear if the smaller SEVI fibrillar species produced in the presence of clusterin (Figure 2f) are cytotoxic or if smaller SEVI oligomers not visible by TEM are cytotoxic. However, these results are consistent with the findings of Narayan et al. [31] who showed that amyloid β 1–40 (Aβ_1–40_) fibril disaggregation in the presence of clusterin results in binding of Aβ_1–40_ oligomers released from fibrils with clusterin to form stable clusterin-oligomer complexes that are toxic to cells. The addition of clusterin alone to the cell culture did not have significant effect on cell viability, consistent with previous studies [31]. Thus, clusterin has a significant protective effect on the cytotoxicity of SEVI towards PC-12 cells. The effect is concentration dependent, with higher concentrations of clusterin (at a sub-stoichiometric level) offering greater protection.

## 4. Discussion

From the results presented in this study, clusterin potently inhibits SEVI amyloid fibril formation in vitro, reduces and prevents the cytotoxicity of SEVI to PC-12 cells, and acts on preformed SEVI fibrils to induce their dissociation. With reference to the first point, SEVI fibrils are present in fresh ejaculate [32,33] implying that clusterin chaperone action does not inhibit SEVI fibril formation completely in vivo. The presence of SEVI in semen implies that SEVI has a functional role under normal physiological conditions, and that SEVI fibril production and localization are regulated tightly in vivo to minimize its potential cytotoxicity. With reference to the third point, the probably oligomeric SEVI species (with clusterin bound) cause cytotoxicity at a comparable level to that of preformed SEVI fibrils (Figure 3). The clusterin-induced dissociation of SEVI fibrils may facilitate the time-dependent degradation of semen amyloid fibrils into smaller peptide fragments via protease action [34]. In semen, this protease-induced degradation of SEVI and other semen amyloid fibrils (e.g., those derived from semenogelin proteins) is most likely responsible for negating the cytotoxicity and HIV-binding capability of the oligomeric SEVI species. Coupled with this, other extracellular chaperones [7,8,9,10,11,12] could associate with the oligomeric SEVI species, for example to prevent their binding to HIV, as has been observed with Hsp104 via association into amorphous-type aggregates [33,35], and when 200 µg/mL of clusterin was incubated with 2 mg/mL PAP248–286 to produce amorphous aggregates (Figure 2e). Furthermore, in semen, efficient extracellular mechanisms are likely to exist to dispose of clusterin-SEVI and other chaperone-SEVI complexes such as macrophage uptake of chaperone-SEVI complexes followed by their intracellular lysosomal degradation [15].

To our knowledge, clusterin is the first extracellular protein (in this case a molecular chaperone) which has been shown to inhibit SEVI fibril formation. As with many other aggregating target proteins [5,6], clusterin is very efficient at preventing the aggregation of PAP248–286 in vitro. In addition to clusterin, a diversity of smaller, non-protein natural products and synthetic molecules inhibit or modulate SEVI fibril formation in vitro and affect HIV infectivity in cell culture. They include: (i) polyhydroxylated antioxidants, i.e., phenols (gallic acid [36], (-)-epigallocatechin-3-gallate (EGCG) [37] and myricetin [38]) and ascorbic acid [39], (ii) polyanions (ADS-J1 [40]), and (iii) hydrophobic nanoparticles [41]. They interact with PAP248–286 and SEVI fibrils via hydrogen bonding, electrostatic or hydrophobic interactions, or combinations thereof. From NMR spectroscopic experiments, the interaction of EGCG with PAP248–286 has been localised to K251-R257 and N269-I277 via mainly positively charged lysine residues [37]. From molecular modelling and molecular dynamics calculations, the binding site of myricetin, ascorbic acid and ADS-J1 to PAP248–286 has been localised to the central fibril-forming region of the peptide [38,39,40]. Furthermore, ADS-J1 disaggregates SEVI fibrils [40], and myricetin and hydrophobic nanoparticles remodel SEVI fibrils via reduction in their β-sheet content [38,41]. Myricetin, ascorbic acid and hydrophobic nanoparticles decrease semen-mediated enhancement of HIV infection in cell culture assays [38,39,41]. The generic ability of hydroxylated small molecules, including polyphenolics, to inhibit amyloid fibril formation, and reduce the associated cell toxicity of peptides and proteins, has been demonstrated in many studies, for example to inhibit amyloid fibril formation of α-synuclein and κ-casein [42,43,44,45]. On the basis of the similarity in the manner of inhibition of SEVI fibril formation by these small, non-protein molecules and clusterin, it is feasible that clusterin interacts with PAP248–286 similarly, including with the central region of the peptide, to inhibit its amyloid fibril formation.

Since clusterin has broad specificity and is a potent extracellular chaperone [5,13,31], it is likely that clusterin will inhibit amyloid fibril formation of other fibril-forming species in semen that enhance HIV infectivity, for example SEM peptides derived from semenogelin proteins [34]. On the basis of the above discussion, it is reasonable to propose that clusterin, on its own or in concert with other extracellular chaperones and/or small molecule inhibitors of SEVI fibril formation, could be utilized therapeutically to prevent HIV infection. 

PAP248–286 has a positive charge of +6 at neutral pH which results in a very high pI value of 10.2 for the peptide [33]. The positive charge is localised to the N- and C-terminal regions of the unstructured peptide, i.e., G248-R257: +4 and K272-Y286: +3, which is slightly modulated by the negative charge at E266 within the peptide’s central region. The central region of PAP248–286, G260-H270, encompasses the amyloid fibril-forming region of the peptide [21]. The presence of fibril-forming regions in the central region of the amino acid sequence of fibril-forming peptides and proteins is common, particularly for those that are unstructured or intrinsically disordered [46]. It has been proposed that the presence of unstructured, dynamic flanking regions (which often are charged, or at least very polar) provides protection for the peptide from fibril formation by inhibiting contact of the central region with the same region of other peptides, and therefore the potential for co-association leading to oligomer formation [46]. 

Furthermore, the high positive charge on SEVI fibrils may be utilized in an antibiotic role in semen. SEVI fibrils are proposed to have an indirect antibiotic activity within the female reproductive tract by binding to the negatively charged lipid surfaces of bacteria in a charge-dependent manner to enable their degradation by macrophages via phagocytosis [33]. Other fibril-forming peptides, including those present in semen, have antibiotic activity [33]. Some of these are well-characterised antimicrobial peptides [47,48,49,50]. The putative antibiotic activity of SEVI implies that it operates as a functional amyloid in semen [51]. Future studies will investigate this possibility and the role of clusterin and other extracellular molecular chaperones in regulating PAP248–286 aggregation and SEVI fibril formation.

## Figures and Tables

**Figure 1 cells-11-03259-f001:**
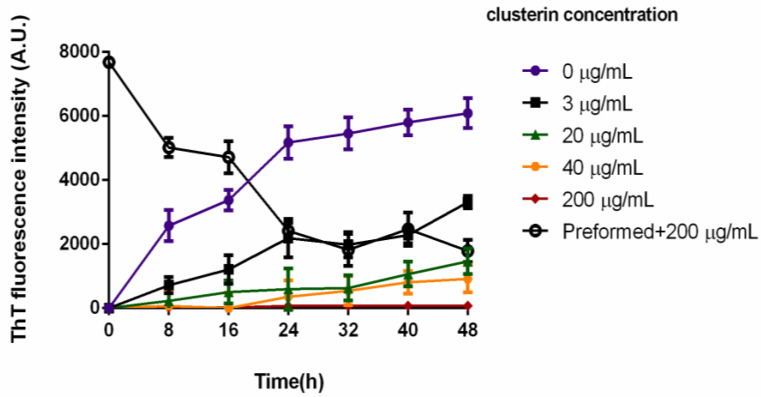
Amyloid fibril formation of 2 mg/mL PAP248–286 peptide (439 μM) with varying concentrations of clusterin (0.050, 0.333, 0.666 and 3.33 μM clusterin monomer, and hence molar ratios of PAP248–286:clusterin monomer of 8788:1, 1318:1, 659:1 and 132:1 respectively) upon incubation for two days at 37 °C and pH 7.2. Also plotted are the results of an experiment in which 2 mg/mL solution of preformed SEVI fibrils was incubated under the same conditions with 200 µg/mL clusterin (3.33 μM). The data shown are means ± SD of triplicates. Some error bars are smaller than the size of the individual data points. Clusterin does not bind ThT in its native state, nor does it form amyloid fibrils in vitro under physiological conditions, and over the time course of this experiment [29].

**Figure 2 cells-11-03259-f002:**
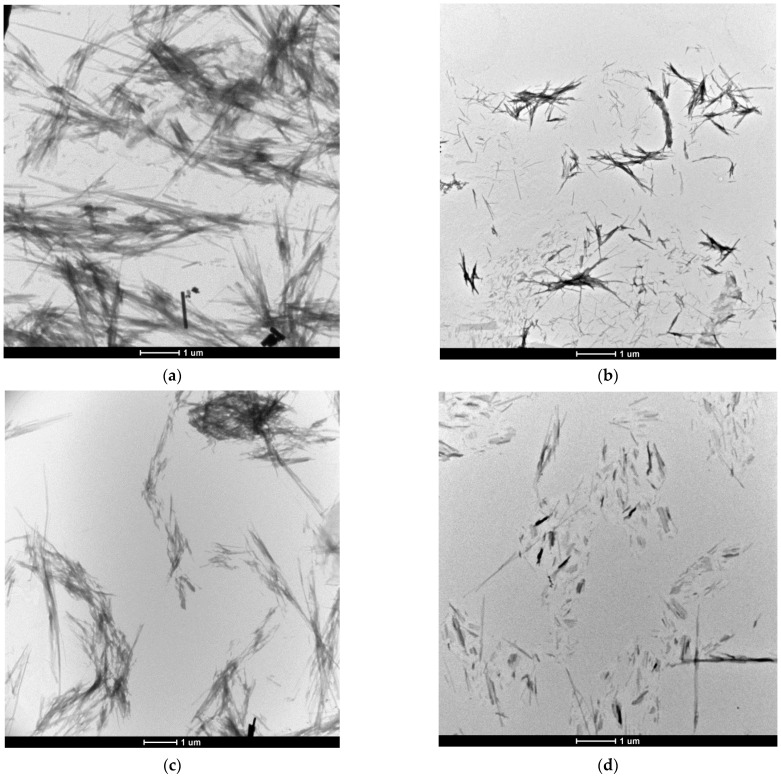
TEM micrographs of the effect of different concentrations of clusterin on SEVI fibril formation. Samples were taken at the end of a two-day incubation with either PAP248–286 or preformed fibrils of SEVI. PAP248–286 was at 2 mg/mL with varying concentrations of clusterin: (**a**) 0, (**b**) 3, (**c**) 20, (**d**) 40, (**e**) 200 µg/mL; (**f**) Preformed 2 mg/mL SEVI fibrils with 200 µg/mL of added clusterin; (**g**) Preformed 2 mg/mL SEVI fibrils PAP248–286 incubated for three days without clusterin. The results are representative of three or more individual experiments. The scale bars represent 1 µm.

**Figure 3 cells-11-03259-f003:**
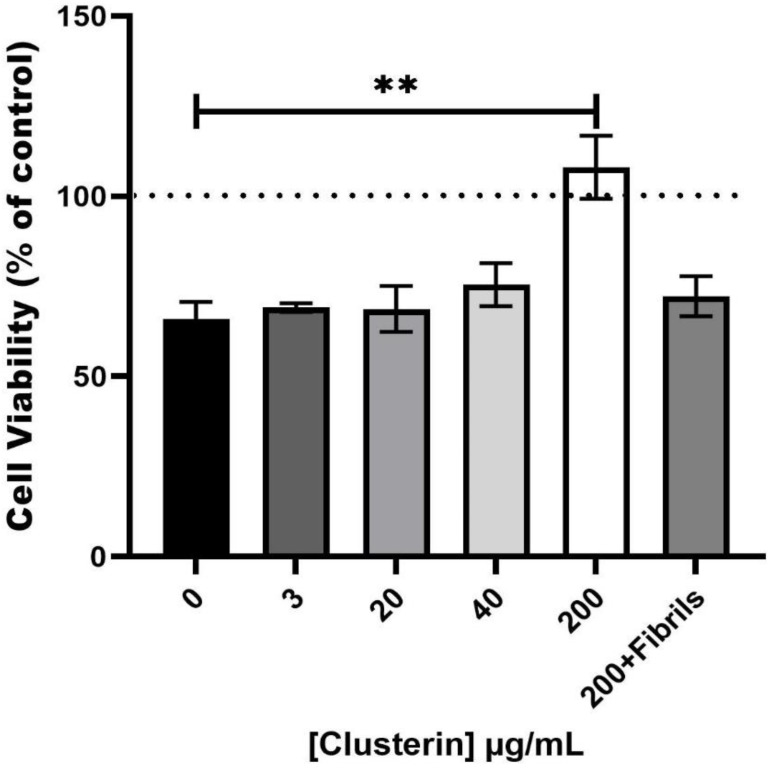
Effect of clusterin on the toxicity towards PC-12 cells of 2 mg/mL (439 μM) PAP248–286. The peptide was dissolved in 200 mM phosphate buffer, pH 7 and incubated at 37 °C for 48 h with different concentrations of clusterin (3, 20, 40 and 200 μg/mL corresponding to 0.050, 0.333, 0.666 and 3.33 μM, and hence molar ratios of PAP248–286:clusterin monomer of 8788:1, 1318:1, 659:1 and 132:1 respectively). In a separate experiment, 200 µg/mL of clusterin was added to preformed SEVI fibrils and incubated for 48 h (200+Fibrils). Samples were then added to the culture media of PC-12 cells and incubated for 48 h. The MTT assay was used to assess cell viability and values are presented as percentage of cell survival compared to control. Results are expressed as mean ± standard error of three independent experiments. (** One-way ANOVA, *p* < 0.01).

## Data Availability

The data presented in this study are available on request from the corresponding author.

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
