# Peer review of "The Extracellular Molecular Chaperone Clusterin Inhibits Amyloid Fibril Formation and Suppresses Cytotoxicity Associated with Semen-Derived Enhancer of Virus Infection (SEVI)"

_cells, 2022, doi:10.3390/cells11203259_

Round 1
Reviewer 1 Report (New Reviewer)
This manuscript presents new information on the role of clusterin, an abundant protein in semen, in blocking amyloid fibril formation derived from an aggregating fragment of prostatic acid phosphatase. The authors provide convincing kinetic spectrophotometric and EM evidence that clusterin - at concentrations known to exist in semen- effectively blocks the fibrillation of this fragment. Further, clusterin was shown to inhibit the cytotoxicity of this fragment in a cell based system. Specific comments on the data and text include the following:
1. Re Figure 1- The molar ratios of clusterin to SEVI should be indicated.
2. Re Figure 2 -The observation that clusterin can dissociate pre-formed fibrils (the 200 ug/ml clusterin sample with pre-formed fibrils) is an unusual finding that requires additional controls. Was any control protein tested at the same time at the same concentration as clusterin? For this sample, it could also be important to know if the pre-formed fibrils lost turbidity as well as ThT binding capability over the 48 h incubation period in the presence of clusterin but not of another protein-- thus providing additional quantitative data to support the idea that clusterin can disaggregate the pre-formed fibrils. (I presume the authors have ruled out clusterin-mediated precipitation of pre-formed fibers on the bottom of the well?)
3. Re Figure 3 (PC12 protection from SEVI cytotoxicity,) I calculate that the concentration of the toxic peptide used in this experiment is over 500 uM, which seems extremely high (but may in fact relate to its actual concentrations in semen-? this should be specifically discussed). Interestingly, the concentration of clusterin used appears to be only in the micromolar range; thus it exhibits a very high effective stoichiometric ratio. Please give molar concentrations and ratios to emphasize this point.
4. The last sentence of the paper does not seem to logically follow from the preceding two sentences.
1) SEVI is a fibril-forming peptide and has antibiotic activity which may (or may not) be due to charge-dependent binding to bacteria. 2) Other fibril-forming peptides also have antibiotic activity.
These two findings do not logically lead to the conclusion that “due to its antibiotic activity”, SEVI operates as a functional amyloid. This should be rephrased to more clearly indicate that the authors feel (I think!) that the antibiotic activity of semen fibril-forming amyloids may represent their actual physiological function. I note that this speculation was not actually explored in the current ms and thus represents an area for future research.
Author Response
We thanks the reviewer for their thoughtful comments and suggestions:
This manuscript presents new information on the role of clusterin, an abundant protein in semen, in blocking amyloid fibril formation derived from an aggregating fragment of prostatic acid phosphatase. The authors provide convincing kinetic spectrophotometric and EM evidence that clusterin - at concentrations known to exist in semen- effectively blocks the fibrillation of this fragment. Further, clusterin was shown to inhibit the cytotoxicity of this fragment in a cell based system. Specific comments on the data and text include the following:
- Re Figure 1- The molar ratios of clusterin to SEVI should be indicated.
Response: These molar ratios have now been included in the caption of Figure 1.
- Re Figure 2 -The observation that clusterin can dissociate pre-formed fibrils (the 200 ug/ml clusterin sample with pre-formed fibrils) is an unusual finding that requires additional controls. Was any control protein tested at the same time at the same concentration as clusterin? For this sample, it could also be important to know if the pre-formed fibrils lost turbidity as well as ThT binding capability over the 48 h incubation period in the presence of clusterin but not of another protein-- thus providing additional quantitative data to support the idea that clusterin can disaggregate the pre-formed fibrils. (I presume the authors have ruled out clusterin-mediated precipitation of pre-formed fibers on the bottom of the well?)
Response:
Clusterin does bind to and stabilise Abeta oligomers that can in turn affect the rate of spontaneous dissociation of the fibrils (see Fig 3 and accompanying Results text in Narayan P, et al,. The extracellular chaperone clusterin sequesters oligomeric forms of the amyloid-β(1-40) peptide. Nat Struct Mol Biol. 2011 Dec 18;19(1):79-83. https://pubmed.ncbi.nlm.nih.gov/22179788/). The electron micrographs in Figure 3 (especially 3F) is consistent with dissociation.
- Re Figure 3 (PC12 protection from SEVI cytotoxicity,) I calculate that the concentration of the toxic peptide used in this experiment is over 500 uM, which seems extremely high (but may in fact relate to its actual concentrations in semen-? this should be specifically discussed). Interestingly, the concentration of clusterin used appears to be only in the micromolar range; thus it exhibits a very high effective stoichiometric ratio. Please give molar concentrations and ratios to emphasize this point.
Response: Yes, the concentration of PAP248-286 (mass 4,551.7 Da) is high (439 uM) relative to that of clusterin (0.050 to 3.33 uM), i.e. the same as that for the in vitro chaperone experiments in Figure 1. The results in Figure 3 emphasise the impressive efficiency of clusterin in inhibiting PAP248-286 aggregation in a cellular environment. The molarity and molar ratios of both species have been included in the caption to Figure 3.
- The last sentence of the paper does not seem to logically follow from the preceding two sentences.
1) SEVI is a fibril-forming peptide and has antibiotic activity which may (or may not) be due to charge-dependent binding to bacteria. 2) Other fibril-forming peptides also have antibiotic activity.
These two findings do not logically lead to the conclusion that “due to its antibiotic activity”, SEVI operates as a functional amyloid. This should be rephrased to more clearly indicate that the authors feel (I think!) that the antibiotic activity of semen fibril-forming amyloids may represent their actual physiological function. I note that this speculation was not actually explored in the current ms and thus represents an area for future research.
Response: Thank you for this suggestion. We have altered the text at the end of the Discussion to state that ‘The putative antibiotic activity of SEVI implies that it acts as a functional amyloid in semen [51]. Future studies will investigate this possibility and the role of clusterin and other extracellular molecular chaperones in regulating PAP248-286 aggregation and SEVI fibril formation’.
Reviewer 2 Report (New Reviewer)
The manuscript by Elias and coworkers investigates the interaction between the extracellular chaperone clusterin and the amyloid-forming fragment of prostatic acid phosphatase known as SEVI (Semen-derived Enhancer of Virus Infection). The authors convincingly demonstrate that clusterin is capable of preventing quite efficiently the onset of SEVI fibrils and, depending on the concentration, sensibly reducing the size of preformed SEVI fibrils. The discussion is very rich of interesting arguments, although at some point a bit speculative, e.g. concerning the clearance of the amorphous heterogeneous aggregates between clusterin and small-size fibrils deriving from the “processing” of large amyloid fibers. Overall I find this manuscript very interesting, well written and worth of publication.
There are a few minor points the authors should correct.
- Page 4, line 170: “SEVI formed in the ...” means probably “SEVI aggregation occurring in the ....”
- Page 4, line 173: “not fibrils” should be “no fibrils”
- Page 6, Figure 2f heading: “Preformed SEVI” should be “Preformed SEVI fibrils”
- Page 7, line 228: : the statement “ ... clusterin prevents fibril formation ...” is too drastic, based on the experimental evidence. Total prevention requires a very large concentration, whereas at low or medium concentrations the prevention is partial. Hence I would suggest to change the statement into “ ... clusterin largely prevents fibril formation ...”
- Page 7, line 243: “...31±6, respectively ...” should read “...31±6 reduction, respectively ...”
- Page 8, line 278: “ ...SEVI production and ...” should read “ ...SEVI fibril production and ...”
- Page 8, line 281: “ ...of SEVI may facilitate ..” should be “ ...of SEVI fibrils may facilitate ..”
Author Response
We thank the reviewer for their thoughtful suggestions
The manuscript by Elias and coworkers investigates the interaction between the extracellular chaperone clusterin and the amyloid-forming fragment of prostatic acid phosphatase known as SEVI (Semen-derived Enhancer of Virus Infection). The authors convincingly demonstrate that clusterin is capable of preventing quite efficiently the onset of SEVI fibrils and, depending on the concentration, sensibly reducing the size of preformed SEVI fibrils. The discussion is very rich of interesting arguments, although at some point a bit speculative, e.g. concerning the clearance of the amorphous heterogeneous aggregates between clusterin and small-size fibrils deriving from the “processing” of large amyloid fibers. Overall I find this manuscript very interesting, well written and worth of publication.
There are a few minor points the authors should correct.
- Page 4, line 170: “SEVI formed in the ...” means probably “SEVI aggregation occurring in the ....”
Response: We prefer to leave the text as is, since it is PAP248-286 that aggregates to produce SEVI fibrils, i.e. SEVI aggregation is not a truly accurate description of what is occurring.
- Page 4, line 173: “not fibrils” should be “no fibrils”
Response: We prefer to leave the text as is, because to change as suggested will alter the meaning of the text from what is intended.
- Page 6, Figure 2f heading: “Preformed SEVI” should be “Preformed SEVI fibrils”
Response: Agreed, this change has been made.
- Page 7, line 228: : the statement “ ... clusterin prevents fibril formation ...” is too drastic, based on the experimental evidence. Total prevention requires a very large concentration, whereas at low or medium concentrations the prevention is partial. Hence I would suggest to change the statement into “ ... clusterin largely prevents fibril formation ...”
Response: We prefer to retain this text as clusterin at the highest concentration (200 ug/mL or 3.33uM) totally inhibits 2 mg/mL (439 uM) PAP248-286 aggregation in vitro and cellular toxicity (Figures 1-3), i.e. at a ratio of 132:1 PAP248-286:clusterin monomer. In response to Reviewer 1’s comments on this, we have included additional text stating the stoichiometric ratios of PAP248-286:clusterin monomer in these experiments which highlights the impressive efficiency of clusterin in inhibiting PAP248-286 aggregation and SEVI fibril formation.
- Page 7, line 243: “...31±6, respectively ...” should read “...31±6 reduction, respectively ...”
Response: Agreed, this change has been made.
- Page 8, line 278: “ ...SEVI production and ...” should read “ ...SEVI fibril production and ...”
Response: Agreed, this change has been made.
- Page 8, line 281: “ ...of SEVI may facilitate ..” should be “ ...of SEVI fibrils may facilitate ..”
Response: Agreed, this change has been made.
Round 2
Reviewer 1 Report (New Reviewer)
The authors have responded to my (minor) concerns; I have no further comments.
This manuscript is a resubmission of an earlier submission. The following is a list of the peer review reports and author responses from that submission.
Round 1
Reviewer 1 Report
In this study, the authors address the possibility of CLUSTERIN inhibiting amyloid fibril formation of SEVI and suppressing its associated cytotoxicity. It had previously been shown that Clusterin may act either as a suppressor or enhancer of amyloid fibrils-inducing pathology. This manuscript did not provide sufficient evidence confirming the role of Clusterin in SEVI inducing enhancement of viral infection or cytotoxicity. Exactly, the manuscript provide limited physiological relevance of their findings.
Major points:
- The manuscript only applied simple methods (Thioflavin T staining and TEM) to illustrate the inhibitory effect of Clusterin on SEVI fibrils formation. In ThT assay, the authors should perform a Clusterin control to exclude the possibility that Clusterin sequester the ThT fluorescence due to the non-specific binding to ThT. Moreover, they should provide more evidence to support their conclusion, such as Congo red staining, CD, PAGE etc.
- The author stated that Clusterin presents at 0.5-3.5mg/mL in semen. However, the concentration of Clusterin tested in the manuscript is not relevant to this physiological concentration (0-200 μg/mL). What will happen when Clusterin tested in higher concentration?
- Lack of any mechanistic study to investigate how Clusterin affects the amyloid-forming process of SEVI and the dissembling process, both of which sharing similar mechanism or not?
- The manuscript only tested PAP248-286, but not SEM peptides or semen samples.
- The most studied function of SEVI is the enhancement of viral sexual transmission. But why choose the cytotoxicity of SEVI on neuronal cells as the observed function? They are totally un-relevant.
- Based on the cytotoxicity results, the protective or evil roles of Clusterin in SEVI fibrils seems more complicated. The author should perform sufficient experiments to further clearly support their conclusion.
Author Response
We thank the reviewer for their thoughtful comments, we have addressed than as follows.
===================================================
- In this study, the authors address the possibility of CLUSTERIN inhibiting amyloid fibril formation of SEVI and suppressing its associated cytotoxicity. It had previously been shown that Clusterin may act either as a suppressor or enhancer of amyloid fibrils-inducing pathology.
All studies (mostly from the laboratory of one of this manuscript’s authors, Prof. Mark Wilson) have shown that clusterin is a very potent molecular chaperone at inhibiting both amorphous and amyloid fibrillar forms of protein aggregation. In a few specific cases, and only at very low sub-stoichiometric ratios of clusterin: amyloid-forming protein (1:50-1:500, depending on the aggregating protein; see Fig. 1 in Yerbury et al. 2007 FASEB J, https://pubmed.ncbi.nlm.nih.gov/17412999/), clusterin does increase the formation of thioflavin T reactive material. Of eight amyloid-forming proteins tested by Yerbury et al., this effect was ONLY seen for three proteins (calcitonin, α-synuclein and amyloid beta). The other five amyloid-forming proteins did not show this effect and all ratios of clusterin:protein tested (between 1:500-1:5) inhibited protein aggregation (a 1:500 ratio of Clusterin:SH3 was insufficient to inhibit its aggregation but a 1:100 ratio certainly did).
So, unless the amount of clusterin available is severely limited compared to aggregating calcitonin, α-synuclein and amyloid beta, then clusterin will certainly inhibit amyloid formation.
- This manuscript did not provide sufficient evidence confirming the role of Clusterin in SEVI inducing enhancement of viral infection or cytotoxicity. Exactly, the manuscript provide limited physiological relevance of their findings.
There is significant evidence presented in this study for the ability of clusterin to modulate the cytotoxicity of PAP248-286 and SEVI fibrils towards PC-12 cells (Figure 3). The effect of clusterin on infectivity is beyond the scope of this paper because it would require access to a specialised contained facility suitable for work with human pathogenic viruses not currently available to us. Given that anti-fibrillar compounds (such as ascorbic acid; Mohapatra et al., 2021 [36]) reduce the infectivity of PAP248-286/SEVI, we included a brief discussion of this point on page 10 in the original version of the manuscript that has been retained in the revised manuscript.
- The manuscript only applied simple methods (Thioflavin T staining and TEM) to illustrate the inhibitory effect of Clusterin on SEVI fibrils formation.
The methods are the ‘industry standard’ and are most appropriate for monitoring the formation and inhibition of amyloid fibril formation. ThT fluoresces upon binding to beta sheet-rich structures (such as the cross beta-sheet conformation of amyloid fibrils) and TEM images confirm that this corresponds to classic amyloid fibrillar structures. There are hundreds of studies in the literature (including our own, e.g. [27,39,41,42]) that have routinely used these techniques, because of their robustness and reliability, to characterise amyloid fibril formation and its inhibition.
- In ThT assay, the authors should perform a Clusterin control to exclude the possibility that Clusterin sequester the ThT fluorescence due to the non-specific binding to ThT.
Our previous studies have clearly shown that clusterin, itself, does not bind ThT in its native state, nor does it form amyloid fibrils in vitro under the physiological conditions and over the time course used in this study (see, for example, the first study that showed clusterin potently inhibited amyloid fibril formation, in this case of the extracellular protein, apolipoprotein C-II: Hatters et al., 2002 [27]; https://doi.org/10.1046/j.1432-1033.2002.02957.x). In the revised manuscript, a note to this effect has been added to the legend of Figure 1.
The increase in ThT fluorescence upon amyloid formation is due to a spectral shift caused by ThT binding to the protein which has altered its conformation significantly from its native state to form to a cross-beta sheet structure. In this study, both the PAP248-286 peptide and ThT are present in excess of clusterin, PAP248-286 significantly so, favouring binding of ThT to the amyloid-forming species. Non-specific binding of excess ThT to clusterin would result in quenching of fluorescence, but as can be seen from Figure 1, no quenching occurs. (See Biancalana and Koide, 2010 for further details; https://www.ncbi.nlm.nih.gov/pmc/articles/PMC2880406/). Furthermore, the TEM images confirm the ThT results, i.e. that clusterin inhibits PAP248-286 amyloid fibril formation.
- Moreover, they should provide more evidence to support their conclusion, such as Congo red staining, CD, PAGE etc.
Standard SDS-PAGE does not confirm amyloid fibril formation. The TEM and ThT results together confirm amyloid formation by very different but complementary techniques. There are many publications that only show ThT binding and fluorescence and others that also show TEM images. Congo red staining is inferior to ThT staining and its use would add no extra information (Biancalana and Koide, 2010; https://www.ncbi.nlm.nih.gov/pmc/articles/PMC2880406/).
We have published far-UV circular dichroism (CD) spectroscopic studies of PAP248-286 and its peptide fragments (Elias et al., 2014 [21]), which clearly show the conversion of the peptides from an unstructured to beta-sheet secondary structure upon fibril formation, thereby confirming results from the ThT experiments and TEM images. CD studies of fibril-forming peptides and proteins in the presence of chaperones such as clusterin are not very informative because both species contribute to the CD spectra. So, deciphering the effect of inhibition of fibril formation by the chaperone (in this case, clusterin) is problematic because it is not possible to ascribe the CD spectroscopic changes occurring to either the aggregating species or the chaperone, since one, or potentially both of them, will be undergoing structural changes during the course of the experiment.
- The author stated that Clusterin presents at 0.5-3.5mg/mL in semen. However, the concentration of Clusterin tested in the manuscript is not relevant to this physiological concentration (0-200 μg/mL). What will happen when Clusterin tested in higher concentration?
The 0.2 mg/mL concentration is relevant to the lower range of clusterin in semen. As we have stated above, and as is apparent from the data presented in Figure 1, at higher stoichiometric ratios complete inhibition of PAP248-286 amyloid formation by clusterin would be expected based on results with all other amyloid-forming peptides and proteins (e.g. Hatters et al., 2002 [27] and Yerbury et al., 2007; https://pubmed.ncbi.nlm.nih.gov/17412999/). So, use of concentrations greater than 0.2 mg/mL clusterin with PAP248-286 in the ThT experiments would not enable further information to be gleaned about the effectiveness of clusterin as an inhibitor of the peptide’s aggregation than is already obtained from the data presented in Figure 1.
- Lack of any mechanistic study to investigate how Clusterin affects the amyloid-forming process of SEVI and the dissembling process, both of which sharing similar mechanism or not?
This would form the basis for an entirely separate (and major) study and is beyond the scope of the present manuscript. Our previous studies have provided insights into the mechanism of clusterin’s inhibition of amyloid fibril formation, e.g. of amyloid beta (Narayan et al. [29]). As the general features of amyloid fibrillar structure are conserved for all fibril-forming peptides and proteins, similar mechanisms may apply for clusterin’s inhibition of PAP248-286 fibril formation, but without experimental evidence for such, it is inappropriate for us to speculate on the mechanistic aspects of clusterin’s inhibition of SEVI fibril formation. As discussed on page 11 of the manuscript, clusterin may interact with PAP248-286 and SEVI fibrils via the central, fibril-forming region of the peptide, in a similar manner to that of a variety of small molecule inhibitors of PAP248-286 aggregation.
- The manuscript only tested PAP248-286, but not SEM peptides or semen samples.
PAP248-286 is the best characterised, most abundant and most infectious of the identified amyloid fibril-forming peptides in semen, including SEM peptides, which are derived from semenogelin proteins [22, 36]). Therefore, PAP248-286 is the most appropriate peptide to study the effect of clusterin on amyloid fibril formation of semen-derived peptides. Since clusterin has broad specificity (i.e. it is promiscuous) and is a potent extracellular chaperone, it is most likely that it will inhibit amyloid fibril formation of any and all SEM peptides. A comment to this effect has been included in the Discussion on page 11 of the revised manuscript.
Semen is a multi-component fluid. Undertaking studies of clusterin’s ability to prevent amyloid fibril formation of semen would be fraught with difficulties, for example, because of the presence of clusterin (and other molecular chaperones) in semen, which would complicate interpretation of results.
- The most studied function of SEVI is the enhancement of viral sexual transmission. But why choose the cytotoxicity of SEVI on neuronal cells as the observed function? They are totally un-relevant.
We do not have access to a specialised viral laboratory to undertake studies of SEVI enhancement of HIV or viral infection. Neurotoxicity is perhaps the best characterised functional readout of the cytotoxicity of amyloid in general, for example for peptides and proteins such as amyloid beta, kappa-casein and PAP248-286/SEVI. As amyloid formation is crucial to both SEVI’s cytotoxicity and to its enhancement of HIV infectivity (see for example [35,36,38]), neurotoxicity is a relevant and rapid functional readout which does not require the hazards of working with infectious virus in specialised laboratories [21].
- Based on the cytotoxicity results, the protective or evil roles of Clusterin in SEVI fibrils seems more complicated. The author should perform sufficient experiments to further clearly support their conclusion.
Our results, summarised in Figures 1 to 3, showed that clusterin potently inhibits the in vitro fibril formation of PAP248-286 and is also effective at inhibiting the cytotoxicity of PAP248-286 (but not SEVI fibrils) towards PC-12 cells. However, as stated at the start of the Discussion: ‘SEVI fibrils are present in fresh ejaculate [30,31] implying that clusterin chaperone action does not inhibit SEVI fibril formation in vivo’, and that SEVI fibrils may have some functional role in semen. We rationalise this on the basis that clusterin dissociates SEVI fibrils into smaller oligomers that are potentially cytotoxic, as occurs with the interaction between amyloid beta fibrils and clusterin [29]. We hypothesise that the production and degradation of SEVI in semen must be tightly controlled by other chaperones and degradative mechanisms, as occurs with the tight regulation of the production, utilisation, and degradation of functional amyloid in vivo. To explore this in detail is a major experimental undertaking, involving at least a variety of biophysical and cell biological studies, that is beyond the scope of this study.
Reviewer 2 Report
The SEVI peptides have been shown to aggregate to form amyloid aggregates. These aggregates of SEVI have been demonstrated to dramatically enhance HIV infection. Previous biophysical and NMR studies have reported the kinetics of amyloid aggregation, fiber morphology, inhibitory effects of EGCG, structures in membrane mimetics and TFE-water, and fusion properties of SEVI. In this study, the authors report an investigation of the effect of cluterin in the inhibition and toxicity of SEVI's aggregation. ThT, cell toxicity, and TEM measurements are reported. Overall, this is a useful contribution to the field.
The manuscript needs a revision:
1) The limitations of the reported results and the approaches used should be clearly mentioned.
2) The individual traces of ThT assay should be included in the supporting information.
3) The deviation from the sigmoidal behavior observed in Fig.1 should be explained.
4) Concentration of the peptide used should be given in each fig caption.
The purity of the peptide should be mentioned. HPLC and mass-spec data showing the purity should be included in the supporting information.
5) The introduction should include previous biophysical and NMR studies on SEVI peptides.
6) The morphology of the fibers observed in this study should be discussed in the context of previously reported results on SEVI.
7) Can the authors obtain secondary structural information via CD experiments? It would be useful to include the structure and structural changes under various conditions used in this study.
8) Previously reported fusion of lipid vesicles results in a mechanism of membrane fusion via which SEVI enhances HIV viral entry to the host cell should be discussed in the context of the reported cell viability data in Fig.3.
Author Response
We thank the reviewer for their thoughtful comments, they have been addressed as follows.
======================================
1) The limitations of the reported results and the approaches used should be clearly mentioned.
We have described, for the first time, that the major extracellular chaperone clusterin is a potent inhibitor, in vitro, of PAP248-286 amyloid fibril formation to form SEVI, and the cytotoxicity of the peptide. As PAP248-286 and clusterin are both present at significant levels in semen, this observation is of clear physiological relevance and is potentially important in understanding the role of PAP248-286/SEVI in semen and the regulation of the peptide’s enhancement of HIV infectivity. At the beginning of the Discussion in the original version of the manuscript, we have discussed this aspect in some detail, including rationalising the contrasting results of our in vitro and cellular experiments with the observation that SEVI fibrils are present in fresh ejaculate.
2) The individual traces of ThT assay should be included in the supporting information.
Since the ThT fluorescence values were obtained from three individual measurements at the same time points, all the data (i.e. the ThT fluorescence values and their error bars) are contained within Figure 1.
3) The deviation from the sigmoidal behavior observed in Fig.1 should be explained.
Time courses, measured using ThT fluorescence, of amyloid fibril-forming peptides and proteins often show a discernible lag phase at the start of the profile, followed by a rapid linear increase and a final plateau phase, i.e. a sigmoidal plot. Amyloid fibril formation usually occurs via a nucleation-dependent mechanism which involves the generation of the oligomeric nucleus (intermediate) during the lag phase which does not bind ThT, and hence no fluorescence is observed. However, some amyloid fibril-forming species do not aggregate via the standard nucleation-dependent mechanism, for example kappa-casein (Thorn et al. (2005) Biochemistry 44, 17027-17036), and they exhibit no lag phase in their ThT fluorescence profile. In the case of the profiles presented in Figure 1 for PAP248-286 in the absence and presence of clusterin, the lag phase could well be lost in the traces shown in the 0-8 h period when no intermediate measurements are shown.
4) Concentration of the peptide used should be given in each fig caption.
The concentration of the PAP248-286 peptide (2 mg/mL) is given in each figure caption.
The purity of the peptide should be mentioned. HPLC and mass-spec data showing the purity should be included in the supporting information.
In the Materials and Methods section, it is stated that the peptide was purchased commercially from Mimotopes. It was > 95% pure, as determined by HPLC. The peptide’s purity has now been included in the text at the bottom of page 3.
5) The introduction should include previous biophysical and NMR studies on SEVI peptides.
PAP248-286 is 39 amino acids in length. As expected for a peptide of this length, it adopts no preferred secondary or tertiary structure in aqueous solution, i.e. it is intrinsically disordered (sometimes referred to as ‘random coil’). Our previous far-UV circular dichroism spectroscopic studies confirmed this (Figure 3 of Elias et al. [21]), as have other studies. Solution NMR spectroscopic studies of PAP248-286 would not be very informative due to the peptide’s lack of structure, and hence an absence of dispersion in its NMR spectrum. The Introduction on page 2 of the revised manuscript now contains additional text stating that PAP248-286 is unstructured, whereas SEVI adopts the cross-beta sheet conformation that is characteristic of amyloid fibrils.
6) The morphology of the fibers observed in this study should be discussed in the context of previously reported results on SEVI.
The morphology of SEVI fibrils, as assessed by TEM, is very similar to that observed in our previous study of the aggregation of PAP248-286 and its fragments (compare Figure 2 of Elias et al. [21] and Figure 2(a) of this manuscript). This point is now made in the text on page 5 of the revised manuscript.
7) Can the authors obtain secondary structural information via CD experiments? It would be useful to include the structure and structural changes under various conditions used in this study.
We have published far-UV circular dichroism (CD) spectroscopic studies of PAP248-286 and its peptide fragments (Elias et al., 2014 [21]), which clearly show the conversion of the peptides from an unstructured to beta-sheet secondary structure upon fibril formation, thereby confirming results from the ThT experiments and TEM images. CD studies of fibril-forming peptides and proteins in the presence of chaperones such as clusterin are not very informative because both species contribute to the CD spectra. So, deciphering the effect of inhibition of fibril formation by the chaperone (in this case, clusterin) is problematic because it is not possible to ascribe the CD spectroscopic changes occurring to either the aggregating species or the chaperone, since one, or potentially both of them, will be undergoing structural changes during the course of the experiment.
8) Previously reported fusion of lipid vesicles results in a mechanism of membrane fusion via which SEVI enhances HIV viral entry to the host cell should be discussed in the context of the reported cell viability data in Fig.3.
The ability of SEVI to induce the fusion of lipid vesicles is relevant to the fusion of the virus membrane with cell membranes (e.g. https://www.ncbi.nlm.nih.gov/pmc/articles/PMC2770606/); it is not relevant to PC-12 cell toxicity. Amyloid toxicity in PC-12 cells is generally accepted to be due to the generation of reactive oxygen species (https://pubmed.ncbi.nlm.nih.gov/9154234/ https://pubmed.ncbi.nlm.nih.gov/22964500/) possibly mediated by RAGE receptors (https://pubmed.ncbi.nlm.nih.gov/23799541/), not membrane destabilisation. While the mechanism of toxicity in PC-12 cells is different from the fusogenic process, amyloid fibril formation is crucial to both SEVI’s cytotoxicity and to its enhancement of HIV infectivity (see for example [35,36,38]). Neurotoxicity is thus a relevant and rapid functional readout that does not require the hazards of working with an infectious virus in specialised laboratories [21].
Round 2
Reviewer 1 Report
Although the authors answered the question, they only cited references or gave some explanation to the question, which is not enough to improve the quality of paper. Each method has its own advantage and disadvantage. As well, there is false positive or negative results. Different batch of reagent or experimental condition might produce distinct results. Control is very important in each experiment. High quality paper use complementary methods to illustrate their results. Inhibition of amyloid and the mechanism is well-studied area. The paper only presented very simple results. Overall, this manuscript presents follow-up of previous study clusterin inhibits amyloid formation but use not physiologically relevant function.
Reviewer 2 Report
The revised manuscript does not satisfactorily address the comments from this reviewer. In addition, the introduction and discussion need major changes; and including brief discussions in the context of previous studies on SEVI would strengthen the manuscript.